# Effect of Size and Loading of Retinoic Acid in Polyvinyl Butyrate Nanoparticles on Amelioration of Colitis

**DOI:** 10.3390/polym13091472

**Published:** 2021-05-02

**Authors:** Jinting Li, Yunmei Mu, Yiwei Liu, Akihiro Kishimura, Takeshi Mori, Yoshiki Katayama

**Affiliations:** 1Graduate School of Systems Life Sciences, Kyushu University, 744 Motooka, Nishi-ku, Fukuoka 819-0395, Japan; lijinting@mail.cstm.kyushu-u.ac.jp (J.L.); ym.mu0820@gmail.com (Y.M.); kishimura.akihiro.776@m.kyushu-u.ac.jp (A.K.); 2Department of Applied Chemistry, Faculty of Engineering, Kyushu University, 744 Motooka, Nishi-ku, Fukuoka 819-0395, Japan; liu.yiwei.975@m.kyushu-u.ac.jp; 3Center for Future Chemistry, Kyushu University, 744 Motooka, Nishi-ku, Fukuoka 819-0395, Japan; 4International Research Center for Molecular System, Kyushu University, 744 Motooka, Nishi-ku, Fukuoka 819-0395, Japan; 5Department of Biomedical Engineering, Chung Yuan Christian University, 200 Chung Pei Rd., Chung Li 32023, Taiwan

**Keywords:** butyrate, polyvinyl butyrate, nanoparticles, inflammatory bowel diseases, all-trans retinoic acid

## Abstract

Butyrate has been used in the treatment of inflammatory bowel diseases (IBD). However, the controlled release of butyrate has been indicated to be necessary in order to avoid the side effects verified at high concentrations. We previously developed nanoparticles (NPs) of polyvinyl butyrate (PVBu) as an oral butyrate donor for the controlled release of butyrate for the treatment of colitis. To examine the effect of the size of NPs on the therapeutic effect of colitis, here we prepared PVBu NPs with different sizes (100 nm and 200 nm). Both sizes of PVBu NPs significantly suppressed the inflammatory response in macrophages in vitro. PVBu NPs with 200 nm showed better effects on the amelioration of colitis compared with the 100 nm-NPs. We found unexpectedly that 200 nm-NP incorporated with all-trans retinoic acid (ATRA) showed a much better therapeutic effect than those with unloaded 200 nm-NPs, although ATRA alone was reported to worsen the inflammation. The synergistic effect of ATRA with butyrate shows evidence of being a promising approach for IBD treatment.

## 1. Introduction

Inflammatory bowel disease (IBD) is an autoimmune ailment that causes chronic inflammation within the gastrointestinal tract including Crohn’s disease and ulcerative colitis [1]. The prevalence of IBD has increased considerably in many countries, imposing a significant economic burden on society and hygiene systems [2,3]. However, underlying reasons for the increasing IBD prevalence are still lacking [4]. Among the important hypotheses, the impaired mucosal barrier and the dysregulation of the immune system are widely supported [5,6]. Despite the efforts in developing IBD therapies, there is still a lack of effective methods [7].

Butyrate, a fermentation product by intestinal commensal bacteria from the dietary fiber, is the endogenous anti-inflammatory agent which has the ability to maintain intestinal health [8,9]. Butyrate is crucial for the maintenance of intestinal immune homeostasis since it is a source of energy for intestinal epithelial cells [10], a stimulator of mucus secretion [11], an inhibitor of histone deacetylase (HDAc) [12], and a ligand for G-protein coupled receptors (GPCRs) [13]. These roles of butyrate are considered to be a promising anti-inflammatory agent to modulate IBD. However, clinical trials of supplementation of butyrate by enema [14] or oral administration [15] have resulted in unsatisfactory outcomes. This may be due to the side effects of butyrate associated with high concentrations; the high concentration of butyrate inhibits the proliferation of intestinal cells and activates immune cells into the inflammatory state [16,17]. This indicates the inability of the current butyrate supplement form to accurately control the amount of butyrate released in the intestine.

Previously, we developed polyvinyl butyrate nanoparticles (PVBu NPs) as a butyrate donor for oral administration [18]. PVBu NPs did not raise the intestinal concentration of butyrate, while they ameliorated symptoms in colitis model mice. These results indicated the resistance of PVBu NPs to hydrolysis under gastric conditions, but released butyrate after phagocytosis by intestinal cells such as resident macrophages. In the intestinal cells, butyrate was released by cellular enzymes to act as an inhibitor of HDAc to induce anti-inflammatory responses. Taking advantage of the hydrophobicity of PVBu, we incorporated vitamin D_3_ as an inducer of anti-inflammatory response and observed the synergistic effect with PVBu in colitis treatment.

Here we examined the influence of different sized PVBu NPs on the therapeutic effects on colitis model mice. It has been reported that a few hundred nanometer-sized particles are suitable to target inflammatory lesions in colitis [19,20,21]. We also examined the effect of incorporation of all-trans retinoic acid (ATRA) in PVBu NPs on the therapeutic effect. ATRA is known to induce an anti-inflammatory response to macrophages [22,23]. However, the functions of ATRA are context-dependent; it worsens the inflammation under the inflammatory conditions [24]. The combination effect of butyrate with ATRA has not yet been reported.

## 2. Materials and Methods

### 2.1. Materials

Vinyl butyrate (VBu) and ATRA were purchased from Tokyo Chemical Industry Co., Ltd. (Tokyo, Japan) and stored at room temperature and 4 °C, respectively. Methanol, toluene and ascorbic acid were purchased from Wako Pure Chemical Industries, Ltd. (Tokyo, Japan). Pluronic F-127, sodium butyrate (SB) and lipase from porcine pancreas were purchased from Sigma-Aldrich (St. Louis, MO, USA). 3,3′-Dioctadecyloxacarbocyanine perchlorate (DiO) was purchased from Takara-Clontech (Shiga, Japan). Dextran sulfate sodium salt in colitis grade (DSS, molecular weight 36,000–50,000) was purchased from MP Biomedicals (Irvine, CA, USA).

### 2.2. Preparation and Characterization of PVBu NPs with Different Size

We synthesized PVBu based on radical polymerization and PVBu NPs were prepared by an oil-in-water emulsion solvent evaporation method according to the previously reported method [18]. The preparation conditions are summarized in Table 1. Briefly, toluene solution (0.5 mL or 2.0 mL) containing PVBu (0.2 g/mL or 0.4 g/mL) was prepared (in the cases of incorporation of ATRA, ATRA (31 μg/mL) was added into 2.0 mL toluene solution containing PVBu (0.4 g/mL) and DiO (6.7 µg/mL as final conc.)). The solution was mixed with 20 mL Pluronic F-127 (1% or 5%) aqueous solution containing 100 mg ascorbic acid. The mixture was homogenized for 10 min at 14,000 rpm or 12,000 rpm (T25 digital Ultra turrax, IKA, Staufen, Germany) and followed by 10 min or 5 min sonication (20% power, 20 kHz, 20 W) with a probe sonicator (UD-211 (TOMY) equipped with a TP-040 tip) to prepare an oil-in-water emulsion. To obtain PVBu NPs without organic solvent, the toluene was evaporated at 300 rpm stirring speed overnight and stored at 4 °C before use. Diameter, polydispersity index (PDI) and zeta potential of PVBu NPs were determined by using Zeta Sizer Nano Series (Malvern Instrument, Malvern, UK) at 25 °C. The diameter and zeta potential were measured in 10 mM HEPES (pH 7.4).

### 2.3. Determination of ATRA Entrapment Efficiency in NP2-RA

The ATRA content in NP2-RA was quantitated by high performance liquid chromatography (HPLC, HITACHI, Tokyo, Japan) equipped with a SunFire™ Pre C18 column (10 mm × 150 mm × 5 μm). NP2-RA was collected by centrifugation at 220,000× *g* for 30 min then dissolved in methanol after removal of the supernatant. Measurement was performed by injecting 20 μL solution and using methanol-acetonitrile (90:10) as mobile phase at a flow rate of 1.0 mL/min at 25 °C. The UV detector was set for the detection of ATRA at 325 nm. The calibration curve was used for ATRA concentration quantification.

### 2.4. In Vitro Butyrare Release from PVBu NPs

The enzymatic reaction was performed by using the pH-stat method [25]. The released butyrate was quantified by quantitative titration with 20 mM NaOH through an automatic titrator (DKK-TOA Co., Tokyo, Japan). Different amounts of lipase (100 mg, 200 mg, 400 mg) were dispersed in 40 mL of 2.5 mM Tris-HCl buffer (pH 7.4) at 37 °C, adding vinyl butyrate (VBu, 34 mM) or PVBu NPs (34 mM in butyrate conc.) then stirred at 400 rpm with a magnetic stirrer.

### 2.5. Cell Culture

RAW 264.7 macrophages were cultured in complete Dulbecco’s Modified Eagle’s Medium (DMEM) containing 10% fetal bovine serum (FBS), 4.5 mg/mL glucose, 100 U/mL penicillin, 100 U/mL streptomycin, supplemented with 500 µg/mL G418. RAW264.7 macrophages transfected with secreted alkaline phosphatase (SEAP) gene with NF-κB responsive promoter was used in the same condition. The cells were cultured in a humidified atmosphere containing 5% CO_2_ and 95% air at 37 °C.

### 2.6. Cell Uptake Study

RAW264.7 macrophages (2 × 10^5^ cells/mL) were seeded in a 96-well glass surface plate. After 24 h, the medium was replaced with the medium containing NP1-DIO and NP2-DIO (5 mM in butyrate conc.) after washing the cells twice time with DPBS. Cellular uptake of NP1-DIO and NP2-DIO was observed by using fluorescence microscope BZ-X700 (Keyence Co., Itasca, IL, USA) after 24 h incubation.

### 2.7. PVBu NPs Inhibitory Effect on NF-κB-Mediated Macrophage Activation

SEAP transfected RAW 264.7 macrophages incubated with DMEM medium were seeded onto a 96-well plate at a density of 2 × 10^5^ cells/mL for 24 h. SB (0.1–20 mM), NP1 or NP2 (each at 5 or 20 mM in butyrate conc.) dispersed in DMEM medium was added to the cells after twice time washing by DPBS. After incubation for 6 h at 37 °C, lipopolysaccharide (LPS) (20 ng/mL in final conc.) was added and incubated for 18 h incubation. SEAP activity in the medium was measured using a microplate reader at a wavelength of 405 nm by collecting supernatants.

### 2.8. DSS-Induced Colitis Model Study

C57BL/6J mice (5-week-old) were fed AIN93G-formula diet (Oriental Yeast, Tokyo, Japan). Mice were treated with drinking water containing NP1 or NP2 (94 mM in butyrate conc.), NP2-RA (94 mM in butyrate conc., 1.6 μg/mL in ATRA conc.), or SB (94 mM) for 10 days. The mice were treated with 2% *wt*/*v* DSS in drinking water from day 3 to day 9. Daily changes in clinical scores were measured to monitor the colitis progression. A daily clinical score assessment was conducted according to the stool consistency and colonic hemorrhage [26].

### 2.9. Statistical Data Analyses

Variance ANOVA followed by Dunnett’s test was performed to analyze the statistical data between groups. *p* < 0.05 was considered statistically significant.

## 3. Results and Discussion

### 3.1. Preparation of PVBu NPs

NPs with the size of approximately 100 to 200 nm have been reported to be appropriate to accumulate in the inflammatory lesions in colonic tissue [19,20,21]. Thus, we tried to tune the size of PVBu NPs in this range. PVBu NPs were prepared by oil-in-water emulsification followed by evaporation of an organic solvent. The characteristics of obtained PVBu NPs are summarized in Table 2. As shown in Table 1, PVBu NPs in different sizes (100 nm or 200 nm) were achieved by tuning the concentrations of PVBu in the organic phase and Pluronic F127 in the aqueous phase as well as the conditions of homogenization and sonication. Surface charges of PVBu NPs were designed to be neutral by using Pluronic F127 as a stabilizer of the PVBu NPs. Pluronic F127 is a triblock copolymer of polyethylene glycol (PEG) and polypropylene glycol (PPG) with a structure of PEG_100_-PPG_65_-PEG_100_ (number in sub is the degree of polymerization) [27]. Therefore, the surface of PVBu NPs is expected to be coated with PEG chains, which is supposed to be suitable to penetrate the mucus layer in the intestine [28]. The colloidal stability of PVBu NPs in the cell culture medium was evaluated. As shown in Appendix A, the size and PDI were not affected by the presence of medium components, indicating the stabilizing effect of the surface coating Pluronic F127. To incorporate ATRA or fluorophore DIO in PVBu NPs, they were dissolved in the organic phase and prepared in the same method. The preparation condition was optimized to minimize the damaging of chemically unstable ATRA as previously reported [29]. Encapsulation efficiency of intact ATRA was determined to be 42% by HPLC.

The butyrate release from NP1 and NP2 was evaluated by using pancreatic lipase of porcine. We used vinyl butyrate (VBu) as a control. As shown in Figure 1, VBu significantly released butyrate by lipase and the increase of the lipase concentration clearly enhanced the butyrate release. In contrast, the butyrate release from NP1 and NP2 was quite slow, which is consistent with our previous report [18]. The slow release behavior of PVBu NPs may be due to the limited accessibility of the lipase to the hydrophobic PVBu chains. We found that the speed of butyrate release from NP1 was somewhat faster than that from NP2 and may be due to the larger specific surface area of NP1. However, the difference was not so significant.

### 3.2. Inhibitory Effect of PVBu NPs on LPS-Mediated Macrophage Activation

A major cause of colitis is considered to be driven by immune cells such as macrophages and released cytokines [30]. Thus, the suppressive ability of PVBu NPs on the inflammatory response of macrophages was evaluated in vitro. First, we evaluated the cellular uptake of PVBu NPs by RAW264.7 macrophages. As shown in Figure 2a, PVBu NPs of both sizes were actively engulfed by the macrophages.

Then, we investigated the suppressive effect of PVBu NPs on the LPS-induced inflammatory response of macrophages (Figure 2b). Here, we used RAW264.7 transfected with SEAP as a reporter gene for NF-κB signaling. SB, which was used as a control, weakly suppressed inflammatory response at lower concentrations (≤1 mM), whereas high concentrations of SB (≥5 mM) rather increased inflammation. These results suggest the importance of controlling the butyrate concentration for the suppressive effect, which may account for the inferior efficacy in the clinical use of SB. In the case of PVBu NPs, however, significant suppression of the inflammatory response was observed for both sizes of PVBu NPs, even at high concentrations. This result indicated that after endocytic uptake, PVBu NPs were processed by intracellular esterase to release butyrate at appropriate concentrations for the suppression of the inflammatory response. The similar effect of NP1 and NP2 may be consistent with the similar releasing speed observed in Figure 1.

### 3.3. PVBu NPs Attenuated DSS-Induced Colitis

The therapeutic effect of PVBu NPs was examined on DSS-induced colitis model mice by oral application [31]. As shown in Figure 3a, the mice were treated with drinking water containing PVBu NPs or SB (94 mM in butyrate conc.) throughout the experiment. From day 3, DSS was added to the drinking water onwards to induce colitis. The progression of colitis was evaluated by a clinical score based on stool consistency and colonic hemorrhage. As shown in Figure 3b, SB did not show significant suppression of the clinical score. However, PVBu NPs showed the size-dependent therapeutic effect; the effect of NP1 was not significant, while NP2 showed significant suppression in the clinical score. The improved therapeutic effect of NP2 beyond NP1 may be attributed to the superior accumulation ability of NP2 in inflammatory lesions due to the release profile of butyrate (Figure 1) and the suppressive effect of macrophages (Figure 2b) were similar in PVBu NPs with different sizes.

The NP2-RA showed a suppressor effect on the amelioration of the clinical score. This positive effect of NP2-RA was unexpected for us. It is notable that ATRA has a contradictory effect on intestinal health depending on the context. In the healthy condition, anti-inflammatory response is enhanced by ATRA, whereas supplementation of ATRA rather enhanced inflammation for the inflammatory lesions [32]. Thus, the significant anti-inflammatory effect of NP2-RA indicated that butyrate changed the context to exhibit the positive effect of ATRA.

## 4. Conclusions

In this study, we prepared PVBu NPs with different sizes (100 nm and 200 nm) which are reported to be suitable for targeting inflammatory colonic areas. In vitro study showed that PVBu NPs engulfed by macrophages regulated the release of butyrate at appropriate concentrations to suppress the inflammatory response. Then anti-inflammatory effect of PVBu NPs was evaluated in colitis model mice. We observed the superior therapeutic effect of NP2 to NP1, which may be due to the difference in the accumulation efficacy of NP2. Interestingly, we found that loading of ATRA into NP2 (NP2-RA) ameliorated colitis more significantly than non-loaded ones (NP2), although ATRA on its own has been reported to aggravate inflammation. The synergistic effect of butyrate and ATRA may be brought by the immunosuppressive background fostered by butyrate in which ATRA exerts its anti-inflammatory potency.

## Figures and Tables

**Figure 1 polymers-13-01472-f001:**
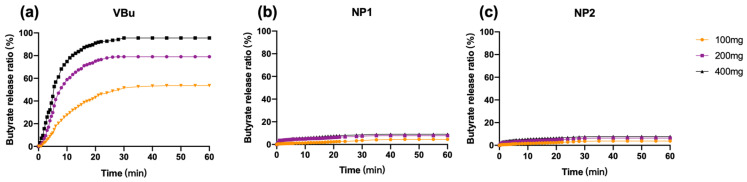
In vitro butyrate release from (**a**) vinyl butyrate (VBu, 34 mM), (**b**) NP1 and (**c**) NP2 (34 mM in butyrate conc.) by pancreatic lipase at 37 °C in 40 mL of 2.5 mM Tris-HCl buffer (pH 7.4) containing 100 mg to 400 mg lipase.

**Figure 2 polymers-13-01472-f002:**
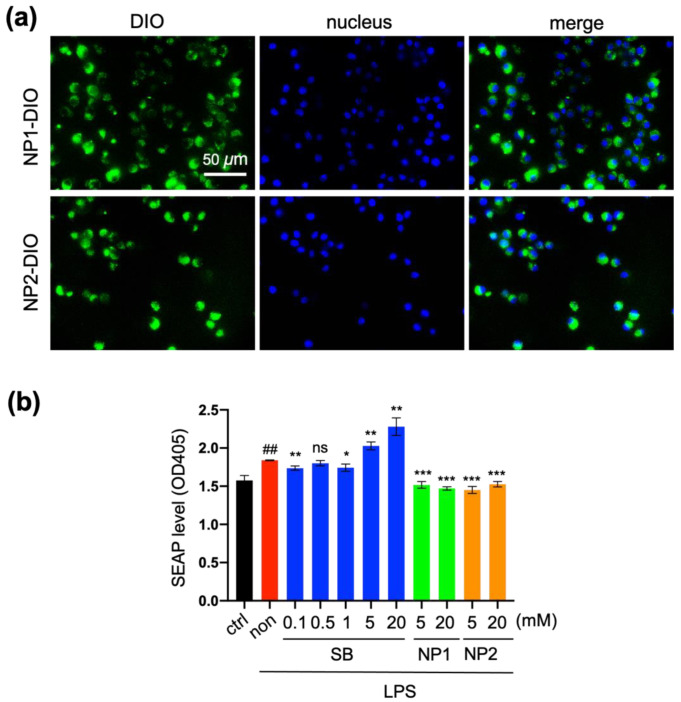
Inhibitory effect of PVBu NPs on NF-κB-mediated macrophage activation. (**a**) Cellular uptake of NP1-DIO and NP2-DIO by RAW264.7 after 24 h incubation. (**b**) Suppressive effect of SB, NP1 and NP2 on the inflammatory response of macrophage. RAW264.7 macrophages were treated with SB or PVBu NPs for 6 h, then stimulated by LPS for 18 h. The SEAP level in the supernatant was measured using a substrate. Data are expressed as mean ± S.D (*n* = 3). ## *p* < 0.01 vs. control; * *p* < 0.05, ** *p* < 0.01, *** *p* < 0.001 and ns vs. non.

**Figure 3 polymers-13-01472-f003:**
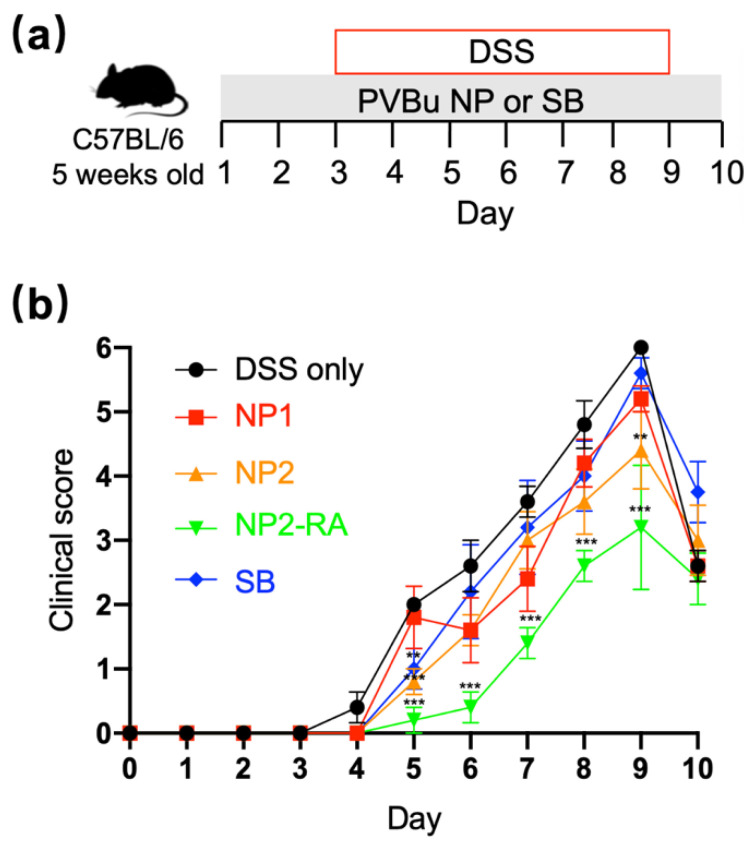
Effect of PVBu NPs on DSS-induced colitis model mice. (**a**) Mice treatment scheme. Mice were treated with drinking water containing PVBu NPs or SB (94 mM in butyrate conc.). 2% *wt*/*v* DSS was added to drinking water from day 3 to 9. During the treatment, the clinical score (**b**) was monitored on a daily basis. Data are presented as mean ± SD (*n* = 5). ** *p* < 0.01, *** *p* < 0.001 vs. control (DSS only).

**Table 1 polymers-13-01472-t001:** Preparation conditions of PVBu NPs.

PVBu NPs Name	PVBu (g/mL)	Toluene (mL)	ATRA (μg/mL)	DiO (μg/mL)	Pluronic F-127 (wt% in 20 mL aq. Phase)	Ascorbic Acid (mg)	Homogenization (rpm)	Sonication (min)
NP1	0.2	0.5	-	-	1	100	14,000	10
NP2	0.4	2	-	-	5	100	12,000	5
NP2-RA	0.4	2	31	-	5	100	12,000	5
NP1-DIO	0.2	0.5	-	6.7	1	100	14,000	10
NP2-DIO	0.4	2	-	6.7	5	100	12,000	5

**Table 2 polymers-13-01472-t002:** PVBu NPs characteristics. Data are expressed as mean ± S.D (*n* = 3).

Pvbu NP Name	Size (nm)	PDI	ζ-Potential (mv)
NP1	119 ± 1.6	0.16 ± 0.02	−0.4 ± 0.10
NP2	205 ± 1.8	0.26 ± 0.03	−0.6 ± 0.14
NP2-RA	206 ± 1.4	0.25 ± 0.02	−0.5 ± 0.03
NP1-DIO	122 ± 0.9	0.16 ± 0.02	−0.4 ± 0.02
NP2-DIO	209 ± 0.7	0.25 ± 0.01	−0.2 ± 0.02

## Data Availability

The data presented in this study are available on request from the corresponding author.

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
