# Peer review of "Effect of Size and Loading of Retinoic Acid in Polyvinyl Butyrate Nanoparticles on Amelioration of Colitis"

_polymers, 2021, doi:10.3390/polym13091472_

Round 1

Reviewer 1 Report

The manuscript describes an interesting piece of work on PVBu NPs, which follows previous studies from the same authors (e.g., ACS Appl Bio Mater 2021, 4, 2335-2341).

In my opinion the manuscript is suitable to publication, provided that the following minor issues are addressed:

-Page 4, line 4: 'evaluation' should be replaced by 'evaporation'.

-Page 4, Table 1: it would be useful (for the reader's convenience) to provide additional information on the manufacturing of a given NPs type, e.g. provide the concentrations of PVBu and/or Pluronic used to obtain NP1, NP2, NP2RA, and so on. 

Author Response

Dear reviewer,

We appreciated all the informative suggestions. Please see the attachment.

Reviewer 2 Report

ABSTRACT

Minor english corrections

  1. “However, controlled release of butyrate has been indicated to be necessary to avoid the side effects at high concentrations of butyrate” Butyrate is repeated in the sentence.

Suggestion: However, the controlled release of butyrate has been indicated to be necessary in order to avoid the side effects verified at high concentrations.

  1. “NPs with 200 nm size showed better effects on the amelioration of colitis than that with 100 nm size.” The word size can be omitted.

Suggestion: “NPs with 200 nm showed better effects on the amelioration of colitis compared with the 100 nm-NPs

  1. “The synergistic effect of ATRA with butyrate will be promising as a therapeutic for IBD treatment”. Reformulation of the sentence.

Suggestion: “The synergistic effect of ATRA with butyrate shows evidence of being a promising approach for the IBD treatment”

INTRODUCTION

  1. “However, the underlying reasons for the process of increasing IBD are still lacking [4]” Reformulation of the sentence.

Suggestion: “However, the underlying reasons for the increasing IBD prevalence are still lacking [4]”

  1. First line of the second paragraph is further in (not aligned with the others).

  1. “This indicates the inability of the current butyrate supplement form to accurately control the amount of butyrate exposed in the intestine”. Other word instead of exposed.

Suggestion: “This indicates the inability of the current butyrate supplement form to accurately control the amount of butyrate delivered (or released) in the intestine”

  1. “Previously, we developed polyvinyl butyrate (PVBu) as a butyrate donor for oral administration [18].” Missing the word nanoparticles.

Suggestion: “Previously, we developed polyvinyl butyrate nanoparticles (PVBu NPs) as a butyrate donor for oral administration [18].”

  1. “These results indicated that PVBu NPs tolerated releasing butyrate by hydrolysis in gastric conditions, while after engulfment by intestinal cells such as resident macrophages, they released butyrate by the cellular enzyme to work as a ligand of HDAc to induce anti-inflammatory response”. The sentence is too long and confusing.

  1. “Utilizing hydrophobicity of PVBu, we incorporated vitamin D3 as an inducer of anti-inflammatory response and observed synergistic effect with PVBu in colitis treatment.” Other word for utilizing.

Suggestion: Taking advantage of the hydrophobicity of PVBu, we incorporated vitamin D3 as an inducer of anti-inflammatory response and observed synergistic effect with PVBu in colitis treatment.”

  1. “Here we examined the effect of the size of PVBu NPs on the therapeutic effects of colitis models.”

Suggestion: “Here we examined the influence of different size PVBu NPs on the therapeutic effects on colitis models”

  1. “It has been reported that a few hundred nanometers are suitable to target inflammatory lesions in colitis [19-21].”

Suggestion: “It has been reported that a few hundred nanometers-sized particles are suitable to target inflammatory lesions in colitis [19-21].”

MATERIALS AND METHODS

  1. “Briefly, toluene solution (0.5 mL or 2.0 mL) containing PVBu (0.2 g/mL or 0.4 g/mL) was prepared [In the cases of incorporation of ATRA, ATRA (31 μg/mL) was added into 2.0 mL toluene solution containing PVBu (0.4 g/mL) and DiO (6.7 μg/mL as final conc)]. The solution was mixed with 20 mL Pluronic F-127 (1% or 5%) aqueous solution containing 100 mg ascorbic acid. The mixture was homogenized for 10 min at 14 000 rpm or 12 000 rpm (T25 digital Ultra turax, IKA, Germany) and followed by 10 min or 5 min sonication (20% power, 20 kHz, 20 W) with a probe sonicator [UD-211 (TOMY) equipped with a TP-040 tip] to prepare an oil-in-water emulsion”. This part of the preparation of the PVBu NPs is confusing, with several conditions being tested. For example, 0.5 mL of toluene was combined with the 0.2 or 0.4 g/mL, or both were tested and mixed with the 0.5 mL of toluene? Did you perform the same for the 2 mL toluene? The same with DiO and ATRA. Confusing for somene to reproduce. Since the authors refer in table 1 NP1, NP2 and so on, you might as well use it here. Or on the other hand, include a table with each condition (example below). 

PVBu NPS

PVBu

(g/mL)

Toluene

(mL)

ATRA

(μg/mL)

DiO

(μg/mL)

Pluronic F-127

(20 mL)

Ascorbic acid (mg)

Homogenization

Sonication

NP1

0.2

0.5

-

?

1%

100

?

?

NP2

0.4

2.0

-

?

5%

100

?

?

NP1-ATRA

0.4

2.0

31

6.7

?

100

?

?

NP1-DiO

?

?

?

?

?

100

?

?

NP2-DiO

?

?

?

?

?

100

?

?

If this is adopted, from this point on, these denominations should be used in figures and text.

  1. “The diameter and zeta potential were measured in water and 10 mM HEPES (pH 7.4), respectively”

Why did the authors choose to measure the size in water and zeta potential in HEPES buffer? The presence of solutes or ions may influence not only the hydrodynamic size of nanoparticles but also the surface charge, by influencing the particle stability in solution. Several are the examples reported in the literature of such influence.  The zeta potential will inform the stability of colloidal dispersion, if the nanoparticles are stable in solution and resist aggregation or, on the other hand, are not stable in solution and will suffer aggregation and flocculate/precipitate. Size and zeta potential need to be performed in the same conditions to be related to each other. If the authors want to present results in both water and HEPES, the zeta potential in water and hydrodynamic size (and PDI)  in HEPES should be performed.

  1. “2.3. Determination of ATRA entrapment efficiency in PVBu NPs.”

Suggestion: “2.3. Determination of ATRA entrapment efficiency in PVBu NPs (NP1-ATRA)”

  1. “Measurement was performed by injecting 20 μL solution and using methanol-acetonitrile (90:10) as mobile phase at a flow rate of 1.0 mL/min.” Missing temperature.

  1. “After 24 h, the medium was replaced with the medium containing DIO loaded PVBu NPs (5 mM in butyrate conc.) in different sizes after washing the cells twice with DPBS.”

Suggestion: “After 24 h, the medium was replaced with the medium containing NP1-DiO and NP2-DiO (5 mM in butyrate conc.) after washing the cells twice with DPBS.”

  1. “The sodium butyrate (SB) (0.1–20 mM) or PVBu NPs (5 or 20 mM in butyrate conc.) with 100nm or 200nm dispersed in DMEM medium”

Suggestion: “The sodium butyrate (SB) (0.1–20 mM) or PVBu NPs (NP1 and NP2, each at 5 and 20 mM in butyrate conc.) dispersed in DMEM medium”

  1. “After incubation for 6 h at 37 °C, lipopolysaccharide (LPS) (final conc. 20 ng/mL) was and incubated for 18 h incubation.” Missing word.

Suggestion: “After incubation for 6 h at 37 °C, lipopolysaccharide (LPS) (final conc. 20 ng/mL) was added and incubated for 18 h incubation”

  1. “100nm or 200nm PVBu NPs (94 mM in butyrate conc.), PVBu NPs containing RA (94 mM butyrate conc., 1.6 μg/mL ATRA conc.), or sodium butyrate (SB) (94 mM) for 10 days.” Use adopted abbreviations/acronyms.

  1. “Table 1.” Size was determined and is the appropriate size reported in the literature to be engulfed by macrophage. And regarding the zeta potential? There is no discussion about the obtained values of zeta potential, its role in nanoparticle stability and its importance for cell uptake. Why was it determined? Include size, pdi and zeta in both water and hepes. Since the nanoparticles were exposed to cells, studies of size and zeta potential in cell culture medium (with and without serum) should be performed for comparison, in order to have a picture of what happens to the particles size and charge under cell culture conditions. Discusse about nanoparticle behavior/zeta potential on the different media.

SECTION 3.2 numbering is repeated (page 4 and 5)

  1. “This result indicated that after endocytic uptake, PVBu NPs were processed by intracellular esterase to release butyrate at appropriate concentrations for the suppression of the inflammatory response.”

In vitro controlled release studies with the PVBu NPs in the presence of esterase (porcine esterase for example) as reported in many other studies in the literature such as in esterase-responsive or esterase stimuli--responsive vesicles or nanoparticles. This should be performed to quantify the butyrate release overtime, in order to provide evidence for this conclusion.

  1. “The improved therapeutic effect of NP2 may be due to the superior accumulation ability of NP2 in inflammatory legions.”

The in vitro controlled release studies with the PVBu NPs in the presence of esterase could reveal size-dependent differences in the release kinetics or amount of butyrate released, influencing the therapeutic effect. The hypothesis of superior accumulation ability of NP2 in inflammatory legions should be supported by tissue accumulation studies.

Author Response

(The authors gave the same response as above.)

Round 2

Reviewer 2 Report

Dear authors, thank you very much for kindly consider the suggested corrections, answer the questions and performing the additional experiments to improve the manuscript understanding and support conclusions. Regards.